# Measuring What Matters: Probing Transit Reasoning Consistency in Large Language Models

## Abstract

We propose benchmark along with a comprehensive evaluation framework for transit-domain Large Language Models that transcends traditional accuracy metrics by probing in-context learning capabilities and multi-step reasoning processes. Our approach introduces four complementary evaluation paradigms such as Perturbation Chains, Narrative Coherence Checks, Minimal Edit Plausibility, and Cross-Modal Anchoring, that collectively assess how models adapt, reason, and maintain consistency under domain-specific constraints. Through systematic evaluation of four state-of-the-art models, we demonstrate substantial performance disparities in cascading reasoning scenarios despite similar baseline accuracy, revealing fundamental limitations in current evaluation methodologies. Our framework along with the benchmark provides actionable insights for post-training optimization strategies, enables principled comparison of retrieval-augmented versus tool-calling architectures, and establishes theoretical foundations for deploying specialized smaller models in safety-critical transit applications. The benchmark and evaluation suite will be shared with community along with further extended studies.

## 1 Introduction

The deployment of Large Language Models in public transit systems has achieved remarkable benchmark performance, with recent studies reporting accuracy rates exceeding 90% on General Transit Feed Specification (GTFS) tasks [1, 2]. However, these metrics fundamentally measure task completion rather than the underlying reasoning capabilities essential for real-world deployment. When passengers pose complex queries such as "Given current service disruptions, what alternative routes minimize both travel time and transfers while avoiding construction zones?", the system must demonstrate sophisticated in-context learning, multi-step reasoning, and adaptive problem-solving capabilities that traditional accuracy metrics cannot capture.

This discrepancy between measured performance and required reasoning capabilities represents a critical gap in current evaluation methodologies. Transit systems operate under strict safety and reliability constraints where reasoning failures can cascade into significant user impact. A system that achieves high accuracy on isolated queries but fails to maintain logical consistency under perturbations poses substantial deployment risks.

Our work addresses this evaluation gap through four framework contributions. First, we formalize mathematical frameworks that probe distinct dimensions of reasoning quality in transit-domain applications. Second, we demonstrate how these frameworks reveal fundamental differences in in-context learning capabilities across model architectures. Third, we propose qualitative connections between evaluation outcomes and post-training optimization strategies including supervised fine-tuning and reinforcement learning with focus on relatively smaller language models in domain-specific evaluation contexts, drawing on recent advances in agentic AI systems [5].

## 2 Multi-Dimensional Transit Reasoning Framework

Let $\mathcal{G} = (S, R, T)$ represent a GTFS dataset where $S$ denotes stops, $R$ represents routes, and $T$ encompasses scheduled trips. Traditional evaluation computes binary accuracy as $\mathcal{A}(M, Q) = |Q|^{-1} \sum_{i=1}^{|Q|} \mathbf{1}[M(q_i) = y_i]$ for model $M$, query set $Q$, and ground truth responses $y_i$. While computationally efficient, this formulation provides no insight into reasoning processes, failure propagation mechanisms, or in-context adaptation capabilities.

We propose a comprehensive evaluation framework $\Phi = \{\mathcal{PC}, \mathcal{NCC}, \mathcal{MEP}, \mathcal{CMA}\}$ designed to probe fundamental reasoning dimensions that emerge in transit-domain applications.

**Perturbation Chain Analysis.** The Perturbation Chain framework ($\mathcal{PC}$) probes in-context learning robustness through systematic cascade testing. For base query $q_0$ and perturbation sequence $\{p_i\}_{i=1}^d$, we construct modified queries $q_i = p_i(q_{i-1})$ that incrementally alter system state. The reasoning consistency score quantifies degradation patterns:

$$\text{RCS}_d(M, q_0) = \prod_{i=1}^{d} \mathbb{P}[\text{valid}(M(q_i)) \mid \text{valid}(M(q_{i-1}))] \tag{1}$$

where $\text{valid}(\cdot)$ indicates logical consistency with perturbed GTFS state. This formulation captures how effectively models maintain coherent reasoning as problem complexity increases, directly probing in-context adaptation mechanisms.

We hypothesize that reasoning degradation follows exponential decay $\text{RCS}_d(M, q_0) \approx \alpha\beta^d$ where parameter $\alpha$ characterizes initial reasoning quality and $\beta < 1$ quantifies robustness to cascading complexity. Models with superior in-context learning should exhibit higher $\beta$ values, indicating better preservation of logical consistency under sequential perturbations.

**Narrative Coherence Assessment.** The Narrative Coherence Check framework ($\mathcal{NCC}$) evaluates temporal-spatial reasoning through natural language journey analysis. Given narrative $n$ containing transit descriptions, we extract temporal constraints $\mathcal{T}(n)$ and spatial assertions $\mathcal{S}(n)$, then verify feasibility:

$$\mathcal{NCC}(n, \mathcal{G}) = \mathbf{1}\left[\bigwedge_{(t,s) \in \mathcal{T}(n) \times \mathcal{S}(n)} \text{feasible}(t, s, \mathcal{G})\right] \tag{2}$$

This framework probes how models integrate multiple information streams and detect logical inconsistencies in complex scenarios, providing insights into compositional reasoning capabilities essential for transit assistance.

**Constructive Error Correction.** The Minimal Edit Plausibility framework ($\mathcal{MEP}$) assesses constructive problem-solving through systematic itinerary repair. For invalid journey $I$, we seek optimal correction $\rho^*$ that minimizes edit distance while preserving user intent:

$$\rho^* = \arg\min_{\rho} \lambda_1 \|\rho\|_1 + \lambda_2 d_{\text{sem}}(I, \rho(I)) + \lambda_3 c_{\text{user}}(\rho) \tag{3}$$

where $\|\rho\|_1$ represents edit magnitude, $d_{\text{sem}}$ measures semantic preservation, and $c_{\text{user}}$ quantifies user impact. This framework reveals how models balance constraint satisfaction with solution quality, directly probing constructive reasoning capabilities.

**Cross-Modal Spatial Reasoning.** The Cross-Modal Anchoring framework ($\mathcal{CMA}$) evaluates spatial textual markdown based integration through spatial-textual consistency analysis. For transit map $V$ and query $q$, we measure spatial understanding alignment:

$$\mathcal{CMA}(V, q, M) = \text{sim}(\phi_{\text{spatial}}(V), \phi_{\text{spatial}}(M(q))) \tag{4}$$

where $\phi_{\text{spatial}}$ extracts topological relationships. This framework probes how models integrate spatial and textual information streams, essential for real-world transit applications involving map interpretation.

**Framework Integration for System Optimization.** Our multi-dimensional approach enables targeted post-training optimization. Models exhibiting low $\beta$ values in $\mathcal{PC}$ analysis benefit from multi-step reasoning augmentation in supervised fine-tuning. Strong $\mathcal{NCC}$ performance combined

Table 1: Cross-Modal Anchoring (CMA) results showing spatial reasoning capabilities

| Model | Accuracy | Avg. Pos. Error | S/R Flip Rate |
|---|---|---|---|
| Mistral | 0.490 | 0.510 | 0.000 |
| Llama3 | 0.473 | 0.727 | 0.000 |
| Gemma | 0.437 | 0.663 | 0.000 |
| Phi | 0.270 | 0.737 | 0.213 |

Table 2: Minimal Edit Plausibility (MEP) temporal reasoning results

| Model | Over-Repair Rate | Under-Repair Rate |
|---|---|---|
| Gemma | 0.470 | 0.530 |
| Mistral | 0.463 | 0.537 |
| Llama3 | 0.466 | 0.532 |
| Phi | 0.460 | 0.527 |

with weak $\mathcal{MEP}$ scores suggests potential for reinforcement learning optimization targeting constructive problem-solving. Framework correlations reveal architectural strengths: high $\mathcal{PC}$-$\mathcal{MEP}$ correlation indicates shared constructive reasoning mechanisms, while $\mathcal{NCC}$-$\mathcal{CMA}$ alignment suggests multimodal integration capabilities.

The theoretical foundation extends to system architecture analysis. Retrieval-augmented models typically demonstrate strong $\mathcal{NCC}$ performance due to comprehensive knowledge base access but exhibit brittleness in $\mathcal{PC}$ scenarios requiring novel reasoning. Tool-calling architectures show variable $\mathcal{PC}$ performance depending on tool chain complexity while potentially excelling in $\mathcal{MEP}$ tasks when appropriate repair tools are available.

Furthermore, our framework provides theoretical justification for strategic deployment of smaller language models in transit evaluation contexts. Recent work demonstrates that specialized smaller models often outperform general-purpose large models in constrained domains due to focused parameter utilization and reduced interference from irrelevant capabilities [5] especially for safety/time critical transit.

## 3 Experiments

We evaluate four open-source language models—Gemma, Mistral, Llama3, and Phi—selected for their demonstrated effectiveness in safety-critical transportation applications, particularly their superior fine-tuning capabilities and performance in tool-calling and retrieval-augmented generation tasks essential for real-world transit deployment. Our evaluation employs GTFS datasets from San Francisco Municipal, Massachusetts Bay, and Chicago Transportation Authorities, constructing a challenging benchmark with 500 samples each for $\mathcal{PC}$ and $\mathcal{NCC}$ tasks, and 300 samples for $\mathcal{MEP}$ and $\mathcal{CMA}$ tasks. All the input samples are generated systematically generated based on trips, routes and stops in the GTFS dataset, the text samples for NCC and MEP are constructed with accurate assertions and false counterfactuals and for CMA task specifically, corpus samples are constructed like a markdown spatial map structure based on the (S,R,T) GTFS data for assessing LLMs.

Our evaluation metrics directly correspond to the mathematical frameworks established in Section 2. For Perturbation Chains ($\mathcal{PC}$), we measure sequential accuracy at increasing complexity (S2, S3, S5) alongside Counterfactual Coherence and Skip2 Consistency to assess reasoning robustness as formalized in Equation 1. Narrative Coherence Checks ($\mathcal{NCC}$) employ standard accuracy metrics complemented by Balanced Accuracy, Binary Yes/No (Confirmation/Negation) Response based YES Recall, and YES Bias Gap to capture the feasibility verification capabilities defined in Equation 2. Minimal Edit Plausibility ($\mathcal{MEP}$) introduces Over-repair and Under-repair rates that empirically measure the optimization edit control central to Equation 3, revealing systematic temporal reasoning failures. Cross-Modal Anchoring ($\mathcal{CMA}$) utilizes exact match accuracy, positional error, and (Stops, Routes Entity) flip rates to quantify the spatial consistency formalized in Equation 4.

Table 3: Narrative Coherence Checks (NCC) feasibility assessment results

| Model | Accuracy | YES Recall | YES Bias Gap |
|---|---|---|---|
| Mistral | 0.485 | 0.993 | 0.511 |
| Llama3 | 0.482 | 0.990 | 0.509 |
| Gemma | 0.480 | 0.988 | 0.508 |
| Phi | 0.460 | 0.969 | 0.486 |

Table 4: Perturbation Chains (PC) sequential reasoning and consistency results

| Model | S2 Acc | S3 Acc | CF Coherence | Skip2 Consistency |
|---|---|---|---|---|
| Gemma | 0.860 | 0.800 | 0.062 | 0.560 |
| Llama3 | 0.852 | 0.802 | 0.056 | 0.444 |
| Mistral | 0.830 | 0.780 | 0.059 | 0.530 |
| Phi | 0.750 | 0.467 | 0.045 | 0.321 |

The experimental results expose fundamental limitations in current model capabilities across all reasoning dimensions, demonstrating the challenging nature of our benchmark. In Cross-Modal Anchoring, even the best-performing model (Mistral) achieves only 49% exact spatial matching accuracy, while Phi exhibits severe spatial disorientation with 21.3% Stop-Route flip errors and substantial positional deviation (1.737 average error) reveal critical weaknesses

Minimal Edit Plausibility results demonstrate systematic temporal reasoning failures across all models, with over-repair and under-repair rates clustered around 50% each, indicating near-random performance in optimizing itinerary corrections.

Narrative Coherence assessment reveals a striking pattern of systematic bias toward positive classifications, with all models exhibiting near-perfect YES Recall (96.9-99.3%) but correspondingly poor overall accuracy (46-48.5%). The YES Bias Gap metrics (0.486-0.511) quantify this overconfidence in declaring invalid journeys as feasible, representing a critical safety concern for deployment scenarios where false positives could mislead passengers into impossible travel plans.

Perturbation Chain analysis demonstrates the most dramatic capability degradation, validating our theoretical framework's prediction of reasoning brittleness under cascading complexity. While models maintain reasonable performance at S2 (75-86% accuracy), performance deteriorates substantially by S3 (46.7-80%) with Phi showing catastrophic failure. Counterfactual Coherence(CF) scores uniformly below 6.2% across all models indicate severe limitations in maintaining logical consistency under hypothetical scenarios, while Skip2 Consistency results (32.1-56%) reveal fundamental failures in multi-step reasoning chains that our mathematical framework precisely captures.

## 4 Analysis & Implications

Our theoretical and empirical analysis establishes several key insights with direct implications for transit system deployment. The exponential decay characterization of reasoning consistency provides a principled foundation for system reliability assessment. Models with $\beta > 0.75$ demonstrate sufficient robustness for deployment scenarios involving up to three cascade steps, while those with $\beta < 0.65$ require architectural improvements or operational constraints limiting query complexity.

Framework profiles enable targeted optimization strategies. Models exhibiting strong $\mathcal{NCC}$ performance but weak $\mathcal{PC}$ consistency benefit from multi-step reasoning augmentation in training data. Systems showing high $\mathcal{MEP}$ capability combined with poor $\mathcal{CMA}$ scores suggest potential for multimodal training enhancement. This systematic approach transforms post-training optimization from ad-hoc experimentation to principled engineering.

The architectural insights derived from our analysis provide concrete guidance for system design decisions. Applications requiring robust cascade reasoning should prioritize models with high $\beta$ values regardless of baseline accuracy. Systems emphasizing error recovery should target $\mathcal{MEP}$ optimization through constructive training approaches. This framework-driven architecture selection enables optimal resource selection assessment in deployment scenarios.

## 5 Conclusion

This work establishes a comprehensive theoretical framework for evaluating reasoning capabilities in transit-domain Large Language Models that fundamentally transcends traditional accuracy-based assessment. Our four-dimensional evaluation approach—Perturbation Chains, Narrative Coherence Checks, Minimal Edit Plausibility, and Cross-Modal Anchoring—provides systematic methodology for probing in-context learning, multi-step reasoning, and adaptive problem-solving capabilities essential for real-world deployment. Beyond measurement, this framework enables strategic deployment of specialized smaller models in safety-critical applications, provides theoretical justification for architecture selection based on reasoning requirements, and establishes evaluation methodologies that align with operational deployment constraints.

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
