# OpenReview forum: "Measuring What Matters: Probing Transit Reasoning Consistency in Large Language Models"
_NeurIPS.cc/2025/Workshop_Mexico_City/NORA — NeurIPS 2025 Workshop NORA Poster_

### Official Review · Reviewer_REr8 · 2025-11-04
**Need to improve by giving more details on evaluations and analysis**

**Rating:** 4
**Confidence:** 3

**Review:**

This paper evaluates the performance of several different LLMs on transit route planning. It evaluates the LLM models based on their reasoning capabilities such as giving the correct next stop after perturbing the routes, and spatial and temporal consistencies. It is a very interesting way to evaluate the reasoning performance of LLMs and is also a very practical application on its own. However I think there are several ways that the paper can be improved:

1. The authors can give some examples on how the datasets and the evaluation tasks look like. Currently the readers can only rely on abstract descriptions and have no ideas on the difficulty level of the tasks.

2. The authors should also give details on what type of prompts they use for the different LLM models in their evaluations. Without these the evaluations are incomplete and are not very helpful to the readers.

3. In the analysis of the performance of the different LLM models, the authors can also give some examples on what type of mistakes they make and why.

Overall I think the authors need to give more details of their work to make it more useful for the readers.

---

### Official Review · Reviewer_zGyS · 2025-11-04
**Well-motivated work, but the technical details are not sufficiently described**

**Rating:** 4
**Confidence:** 3

**Review:**

**Summary**

This paper presents a new evaluation framework for LLM-based transit applications that moves beyond simple accuracy metrics. The authors introduce four evaluation paradigms: Perturbation Chains, Narrative Coherence Checks, Minimal Edit Plausibility, and Cross-Modal Anchoring. Each task aims to assess how well models can adapt to diverse constraints and reason about logical, spatial, and temporal conditions. Four strong LLMs are evaluated using this framework, revealing limitations in handling complex transit scenarios, despite the models achieving similar baseline accuracy. The proposed framework is designed to inform mitigation strategies such as targeted post-training and facilitate principled comparisons between different model architectures.


**Strengths**

1. The paper focuses on an interesting and emerging application area for LLMs in the transit domain, which could inspire workshop attendees.
2. The proposed evaluation framework addresses real-world needs in transit applications, where models must handle complex reasoning under various domain-specific constraints.
3. The high-level design of the four evaluation tasks is reasonable and well-motivated.

**Weaknesses**

1. The technical details of the proposed evaluation framework are insufficient, which makes it very difficult to assess its effectiveness or to reproduce the results. For example, the specific prompts used for each task are not provided, and the implementation of each task is unclear. As the paper does not yet reach the 8-page limit, it would be beneficial to add more detail about the experimental setup.
2. Although the paper attempts a theoretical analysis, it lacks supporting evidence such as formal proofs or empirical results. In particular, the hypothesis presented in Lines 52-55 is not substantiated within the paper.
3. The paper is not self-contained and assumes familiarity with the transit domain. Providing more background information and clearer definitions of key terms would make the discussion more accessible, especially for readers unfamiliar with prior work, such as that by Devunuri et al.

---

### Official Review · Reviewer_QXrh · 2025-11-04
**Review of Measuring What Matters**

**Rating:** 4
**Confidence:** 3

**Review:**

This paper presents a novel evaluation framework for transit-domain Large Language Models, proposing to replace traditional accuracy metrics with a more sensitive system that probes in-context learning capabilities and multi-step reasoning processes.

The basic proposals here seem logically well-founded, but the paper does not demonstrate enough motivation and in some cases, novelty. For example, the idea of question perturbation to test and improve reasoning models is not new: https://arxiv.org/pdf/2107.13935
and that paper went into some detail on the complexities of creating these perturbation tests.

Many parts of the paper are not well explained, and examples of the datasets and scenarios should at least be included in an appendix.

As the descriptions of these metrics currently read, they don't all seem independent? It seems first the system must detect an Invalid journey. Then for those it does correctly, apply the edit distance to optimize the correction. Does MEP depend on NCC? This is unclear. Obviously dependencies in evaluation metrics are to be avoided.

line 99 typo: "...generated systematically generated..." --> "...systematically generated"

Some problem framing would improve this paper. How are the current evaluation measures inaccurate or insufficiently useful for the data consumer?

 For example, it seems that they are positing the following:

1. how effectively models maintain coherent reasoning as problem complexity increases, directly probing in-context adaptation mechanisms.

2. how models integrate multiple information streams and detect logical inconsistencies in complex scenarios

3. how models balance constraint satisfaction with solution quality, directly probing constructive reasoning capabilities.

4. how models integrate spatial and textual information streams, for map interpretation

Some initial introduction of these capabilities with respect to the datasets would be helpful for the reader. Also, an explanation of how the capabilities are distinct. For example test (2) and (4) seem very similar.

That said the results are actually quite interesting, it's just that the paper does not properly set up the expectation that those results should be surprising. I suggest the authors expand this into a longer paper with more examples, problem statement and references.

---

### Official Review · Reviewer_ZRCo · 2025-11-07
**Theoretical paradigms for LLM evaluation**

**Rating:** 8
**Confidence:** 3

**Review:**

Pros:
(1) This paper lays out a framework of four essential paradigms, to probe underlying reasoning capabilities of LLMs that emerge in transit-domain applications. The insights may also be helpful in model optimization, and in identifying the strengths and weaknesses of various types of models. The ideas are reasonable and innovative.
(2) The framework can extend to other domains using LLMs.

Cons:
(1) Because the paper is limited by space, the details of such evaluation paradigms are missing. It is difficult to understand how each method works. It is also unclear what the data looks like.

---

### Official Review · Reviewer_ZRCo · 2025-11-07
**Theoretical paradigms for LLM evaluation**

**Rating:** 8
**Confidence:** 3

**Review:**

Pros:
(1) This paper lays out a framework of four essential paradigms, to probe underlying reasoning capabilities of LLMs that emerge in transit-domain applications. The insights may also be helpful in model optimization, and in identifying the strengths and weaknesses of various types of models. The ideas are reasonable and innovative.
(2) The framework can extend to other domains using LLMs.

Cons:
(1) Because the paper is limited by space, the details of such evaluation paradigms are missing. It is difficult to understand how each method works. It is also unclear what the data looks like.